# Exploration of Cytochrome P450-Related Interactions between Aflatoxin B1 and Tiamulin in Broiler Chickens

**DOI:** 10.3390/toxins16030160

**Published:** 2024-03-20

**Authors:** Pan Sun, Orphélie Lootens, Tadele Kabeta, Diethard Reckelbus, Natalia Furman, Xingyuan Cao, Suxia Zhang, Gunther Antonissen, Siska Croubels, Marthe De Boevre, Sarah De Saeger

**Affiliations:** 1Department of Bioanalysis, Centre of Excellence in Mycotoxicology and Public Health, Faculty of Pharmaceutical Sciences, Ghent University, B-9000 Ghent, Belgium; pan.sun@ugent.be (P.S.); orphelie.lootens@ugent.be (O.L.); marthe.deboevre@ugent.be (M.D.B.); 2Laboratory of Pharmacology and Toxicology, Department of Pathobiology, Pharmacology and Zoological Medicine, Faculty of Veterinary Medicine, Ghent University, B-9820 Merelbeke, Belgium; firanfiri.04@gmail.com (T.K.); diethard.reckelbus@ugent.be (D.R.); natalia.furman@ugent.be (N.F.); siska.croubels@ugent.be (S.C.); 3Department of Veterinary Pharmacology and Toxicology, College of Veterinary Medicine, China Agricultural University, Beijing 100193, China; cxy@cau.edu.cn (X.C.); suxia@cau.edu.cn (S.Z.); 4Laboratory of Medical Biochemistry and Clinical Analysis, Department of Bioanalysis, Ghent University, B-9000 Ghent, Belgium; 5School of Veterinary Medicine, College of Agriculture and Veterinary Medicine, Jimma University, Jimma P.O. Box 307, Oromia, Ethiopia; 6Chair Poultry Health Sciences, Department of Pathobiology, Pharmacology and Zoological Medicine, Faculty of Veterinary Medicine, Ghent University, B-9820 Merelbeke, Belgium; gunther.antonissen@ugent.be

**Keywords:** CYP450 enzymes, tiamulin, aflatoxin B1, broiler chicken, mycotoxin-drug interaction, pharmacokinetics

## Abstract

Poultry may face simultaneous exposure to aflatoxin B1 (AFB1) and tiamulin (TIA), given mycotoxin contamination and antibiotic use. As both mycotoxins and antibiotics can affect cytochrome P450 enzymes (CYP450), our study aimed to explore their interaction. We developed UHPLC-MS/MS methods for the first-time determination of the interaction between TIA and AFB1 in vitro and in vivo in broiler chickens. The inhibition assay showed the half maximal inhibitory concentration (IC_50_) values of AFB1 and TIA in chicken liver microsomes are more than 7.6 μM, indicating an extremely weak inhibitory effect on hepatic enzymes. Nevertheless, the oral TIA pharmacokinetic results indicated that AFB1 significantly increased the area under the plasma concentration-time curve (AUC_last_) of TIA by 167% (*p* < 0.01). Additionally, the oral AFB1 pharmacokinetics revealed that TIA increased the AUC_last_ and mean residence time (MRT) of AFB1 by 194% (*p* < 0.01) and 136%, respectively. These results suggested that the observed inhibition may be influenced by other factors, such as transport. Therefore, it is meaningful to further explore transport and other enzymes, involved in the interaction between AFB1 and TIA. Furthermore, additional clinical studies are necessary to thoroughly assess the safety of co-exposure with mycotoxins and antibiotics.

## 1. Introduction

Mycotoxins are toxic secondary metabolites produced by various types of fungi, including *Alternaria*, *Aspergillus*, *Fusarium*, and *Penicillium*. More than 400 mycotoxins have been identified so far, such as aflatoxins (AFs), ochratoxin A (OTA), fumonisins (FUMs), and zearalenone (ZEN), which are hazardous to human and domestic animals [1,2]. Among them, aflatoxin B1 (AFB1) is known as one of the most toxic mycotoxins due to genotoxic, mutagenic, teratogenic, and carcinogenic effects [3,4]. It is commonly reported that these toxic compounds are frequently detected in poultry feed, constituting a great threat to the health of both animals and humans [2,5,6]. The global surveys revealed the worldwide occurrence of myco-toxins regarding tested samples from Europe, Africa, and Asia, and more seriously, samples were contaminated with at least one mycotoxin [2,7]. AFB1 levels ranged from 0.5–36.4 µg/kg feed, indicating that mycotoxin contamination in China urgently requires more attention and monitoring [8]. Ma et al. also demonstrated that AFB1 is commonly found in feed ingredients and commercial feed in China, ranging from 1.3 to 10 µg/kg [9]. In Spain, AFB1 was found at a maximum concentration of 6.9 μg/kg in poultry feed [10]. More seriously, AFB1 levels were as high as 39.9 μg/kg in Pakistan [11], where 100% of grower broiler feed was contaminated with AFB1 [12]. In Egypt, it was observed that AFB1 and aflatoxin B2 (AFB2) were both present in animal feedstuff ranging from 0.851 to 1.363 µg/kg [13]. Even, AFB1 concentrations of 1067 µg/kg have been reported in poultry feed in West Africa, highlighting the widespread presence of AFs in feed commodities [14]. In sub-Saharan Africa, poultry feeds were co-contaminated with more than 21 mycotoxins and/or metabolites [15]. In the United States, most of the corn samples had co-occurrences of at least two detectable mycotoxin (≥2 mycotoxins) [16]. Similarly, co-occurrence of mycotoxins has also been observed mainly with AFs and FUMs in Latin American countries [17]. AFB1 can induce liver cell damage in poultry through a variety of molecular mechanisms, and the main of damage mechanisms that have been discovered so far include oxidative damage, promoting apoptosis, influencing hepatocyte gene expression, interfering with hepatocyte autophagy, pyroptosis and necroptosis [18]. As reported, it is well documented that AFs are mainly metabolized by CYP450 enzymes [15,19]. In broilers, it was summarized that hepatic CYP1A1, CYP1A2, CYP2A6 and CYP3A4 enzymes are mainly involved in AFB1 biotransformation [20]. The gene expressions of CYP3A4 and CYP3A7 in poultry were influenced [21].

Antibiotics such as sulfonamides, tetracyclines, quinolones, macrolides, aminoglycosides, and β-lactam antibiotics, have been widely used as supplements or as additives in animal feed to promote animal growth and prevent diseases [22,23,24]. Because of the wide usage, a variety of antibiotics were found in livestock products, potentially posing health risks to the public due to antibiotics residues [25]. A study implied that lincomycin inhibits CYP1A, CYP2A, CYP2C, CYP2B, CYP2E and CYP3A in porcine liver microsomes at a concentration of 3 μM [26]. Amoxicillin inhibited CYP2C8 with an IC_50_ of 830 µM by recombinant *Escherichia coli* [27]. Trimethoprim also shows a weak inhibitory effect on CYP2C8 [28,29]. Meanwhile, enrofloxacin was found to inhibit CYP3A activity in chickens at doses of 25 and 125 mg/kg body weight (BW) [30]. Antonissen et al. showed that the pharmacokinetics of CYP1A4 substrate drugs, such as enrofloxacin might be altered by simultaneous FUMs exposure in chickens [31]. In addition, a significant upregulation of CYP1A4 was observed in chickens fed a diet contaminated with FUMs [31]. The traditional medicine curcumin, with its broad spectrum of antibacterial actions, was found to inhibit CYP2A6 enzyme activity at a dose of 450 mg/kg feed in the detoxification of AFB1 [32]. The metabolism of enrofloxacin was affected by tilmicosin via inhibition of CYP3A4 [33]. Besides, tilmicosin increased the residual concentrations of enrofloxacin in broilers. Among the pleuromutilin antibiotics, tiamulin (TIA) has been commonly used in poultry over 30 years, and its use has not largely led to a general increase of resistance to many pathogens [34] TIA administered at therapeutic levels is relatively quickly absorbed, metabolized in the liver, and rapidly eliminated from the chickens’ body. Moreover, it showed a strong interaction with the ionophore antibiotics (coccidiostats) monensin, narasin, and salinomycin at therapeutic levels [34]. In vitro, TIA was an effective inhibitor of CYP450 activities in goat and cattle microsomes [35], where a stable metabolic intermediate complex was formed with TIA. The inhibition phenomenon was also observed in pig liver microsomes [36]. In another study, it was clarified that TIA acted as an inhibitor of porcine hepatic CYP3A [26]. Importantly, TIA also influenced certain CYP450 enzymes in chickens, such as CYP1A, 2A, 2B, 2E, and 3A [37,38]. In summary, TIA was found to inhibit various CYP450 enzyme activities in rats, pigs, chickens, and goats, although species- or dose-related differences were reported [35,36,39].

Importantly, Lootens et al. suggested that drugs are more likely to be perpetrators of CYP450-mediated drug-mycotoxin interactions in animals, while mycotoxins are more likely to be considered the victims, based on the usually higher systemic levels of drugs compared to mycotoxins [40]. Therefore, the assumption has been made that antibiotics could influence the disposition of AFs in poultry. In general, few research has been conducted on the CYP450-related mycotoxin and antibiotics interaction [31]. Therefore, the objective of this study was to determine the interaction of the antibiotic TIA and the mycotoxin AFB1 (Figure 1) in vitro and in vivo in broiler chickens using ultra-high performance liquid chromatography tandem mass spectrometry (UHPLC-MS/MS) and pharmacokinetic methods.

## 2. Results

### 2.1. Validation of the Analytical Methods

The established methods for the six probe substrates, TIA and AFB1 in CLM, and for TIA and AFB1 in chicken plasma were thoroughly validated, including assessing sensitivity, linearity range, accuracy, and precision. In this study, the limits of detection (LODs) for 4-acetaminophen (ACE, ACE is the hydroxylated form of PH), hydroxybupropion (OH-BP), 6-hydroxychlorzoxazone (OH-CLN), 4-hydroxytolbutamide (OH-TBD), 7-hydroxycoumarin (OH-CAN), and 1-hydroxymidazolam (OH-MDZ) were 2 ng/mL in CLM. The obtained limits of quantification (LOQs) for ACE, OH-BP, OH-CLN, OH-TBD, OH-CAN, and OH-MDZ were 5 ng/mL in CLM. The LOQs for AFB1 and TIA analysis in chicken plasma were 0.5 ng/mL and 5 ng/mL, respectively. The LODs for AFB1 and TIA analysis in chicken plasma were 0.2 ng/mL and 2 ng/mL, respectively. The calibration curves for all analytes were linear over the concentration ranges with a correlation coefficient (R^2^) > 0.99. The obtained accuracy and precision of the methods were evaluated using recoveries and coefficient of variation (CV) for intra-day and inter-day measurements. The mean recoveries ranged from 90.4 to 108.6%, with CVs lower than 13.8%, and are summarized in Table 1 and Table 2, which suggested that the methods had suitable accuracy and precision.

### 2.2. Enzyme Kinetics Profile of the Probe Substrates

An enzyme kinetic study was conducted to optimize the probe substrate concentrations. The Michaelis-Menten equation was used to calculate the maximal velocity (V_max_) and Michaelis-Menten constant (K_m_) for PH, BP, CLN, TBD, CAN and MDZ in CLM. The kinetic profiles of six probes are represented in Figure 2. Meanwhile, the optimal incubation time, microsomal protein concentration, and substrate concentration for the probes are listed in Table 3. As shown in Table 3, the K_m_ values of PH, BP, CLN, TBD, CAN, and MDZ were 20.7 μM, 9.4 μM, 21.8 μM, 14.4 μM, 56.1 μM, and 3.3 μM, respectively.

### 2.3. Inhibition Assay of Aflatoxin B1 and Tiamulin on the Activity of CYP450

The determination of IC_50_ values is important to evaluate the potential of compounds to inhibit CYP450 enzymes. In the present study, IC_50_ values of TIA and AFB1 were determined in CLM, as shown in Figure 3 and Figure 4, respectively. AFB1 inhibited CYP2A6 and CYP3A4, with IC_50_ values of 24.1 μM and 41.4 μM, respectively. While IC_50_ values were 147.0 μM, 662.6 μM, 271.2 μM, 526.7 μM for CYP1A2, CYP2B6, CYP2E1 and CYP2C9, respectively. For TIA, IC_50_ values for CYP1A2, CYP2B6, CYP2E1, CYP2C9, CYP2A6, and CYP3A4 were 65.8 μM, 101.9 μM, 47.6 μM, 65.6 μM, 7.6 μM, and 334.2 μM, respectively, suggesting a negligible level of inhibition observed for CYP3A4.

### 2.4. Pharmacokinetics of Aflatoxin B1 and Tiamulin in Chickens

No significant adverse health effects were observed in broilers when AFB1 and TIA were co-administrated. However, it is noteworthy to mention that during the course of the experiment, one bird (n° T5) died in the group that received AFB1 in the feed (on day 8 after start of the trial). The histological examination results indicated that the cause of death was not related to AFB1 exposure. Plasma concentration data (>LOQ) were analyzed using software WinNonlin (v.8.3.4) via noncompartmental modeling to obtain primary pharmacokinetic parameters. An overview of the plasma concentrations of TIA and AFB1 after oral administration is shown in Appendix A. The pharmacokinetic parameters are shown in Table 4 and Table 5. 

The pharmacokinetic profile of TIA is demonstrated in Figure 5A. Liao et al. implicated exposure to airborne AFB1 of humans was a relatively high-risk during livestock feeding and storage bin cleaning [41]. Therefore, a safe AFB1 concentration of 20 μg/kg in feed was used as officially recommended as maximum level by EU [42]. Following oral administration of TIA at a dose of 25 mg/kg BW, the drug was rapidly absorbed in AFB1-unexposed broiler chickens. The plasma drug concentration reached its peak concentration of 89.9 ± 45.9 ng/mL at 0.44 h post administration. As reported, CYP1A and CYP3A protein levels were decreased in piglets exposed to a very high level of AFB1 contamination (1807 μg/kg AFB1 in feed) [43]. For the mean pharmacokinetic parameters of TIA, the elimination half-life time (T_1/2_) and mean residence time (MRT) were 7.0 ± 2.9 h and 6.8 ± 1.4 h in AFB1-unexposed broilers, respectively. The AUC_last_ and peak concentration (C_max_) in the AFB1-uncontaminated group were 294.5 ± 63.3 h·ng/mL and 89.9 ± 45.96 ng/mL, respectively. AFB1 exposure increased the values of AUC_last_ and C_max_ of TIA by 167% and 155%, respectively. Additionally, apparent volume of distribution (V_d_F_obs_) and apparent body clearance (CL__F_obs_) in AFB1-exposed group were 57% and 63% lower than in the control group (*p* < 0.01).

The pharmacokinetic profile of AFB1 is shown in Figure 5B, following oral administration of AFB1 at a dose of 1 mg/kg BW. A noncompartmental model was used for AFB1 plasma concentration description. The obtained AUC_last_ and MRT were 170 ± 103.6 h·ng/mL and 6.4 ± 3.2 h for TIA-exposed chickens, respectively, and 87.5 ± 39.6 h·ng/mL (*p* < 0.01) and 4.7 ± 0.7 h for TIA-unexposed broilers, respectively. The V_d_F_obs_ and CL__F_obs_ values were 18.4 ± 7.7 L/kg and 9.4 ± 6.5 L/h/kg in TIA-unexposed broilers, and 14.8 ± 4.8 L/kg and 14.3 ± 6.7 L/h/kg in TIA-exposed chickens. Table 5 shows that there was no significant difference between the treatment and control groups in terms of T_1/2_ or apparent body clearance (CL). However, there was a significant difference between the groups in terms of C_max_ and the area under the plasma concentration time curve from 0 to the last time point AUC_last_. In addition, as shown in Figure 4B, the curves exhibited a similar trend during the initial phase from 0.5 h to 4 h, which suggested that type of interaction is formation-limited.

## 3. Discussion

Six CYP450 probe substrates, namely PH, BP, CLN, TBD, CAN, and MDZ for CYP1A2, CYP2B6, CYP2E1, CYP2C9, CYP2A6, and CYP3A4, respectively, were used to assess the enzyme kinetics in CLM. The K_m_ and V_max_ parameters showed the presence of enzymatic activity towards the six probe substrates tested and suggested the existence of broiler chicken CYP450 orthologues of mammalian CYP1A2, CYP2B6, CYP2E1, CYP2C9, CYP2A6, and CYP3A4 enzymes [44]. In humans, the incubation times were 60 min for CYP1A2, CYP2A6, CYP2B6, CYP2C19, CYP2E1, and 120 min for CYP3A5 [45]. The incubation times were 5 min for CYP2C9 and CYP3A4, 15 min for CYP1A2, CYP2A6, and CYP3A4, and 20 min for CYP2E1 in present study. 

The CYP450 isoenzymes of CYP1A, CYP2A, CYP2B, CYP2C, CYP2D, CYP2E, and CYP3A were measured by commercially available antibodies in broilers [46]. CYP3A activity was studied in vitro using MDZ as a substrate, showing high CYP3A activity in the livers of broilers [47]. It was pointed out that the enzyme CYP3A37 has the same activity of CYP3A4 in humans [48]. It was summarized that CYP2A6, CYP3A37, CYP1A5, and CYP1A1 were responsible for bioactivation of AFB1 in poultry species [49]. The total CYP450 content in chickens was 0.296 ± 0.04 nmol/mg [50] Meanwhile, the biotransformation activity of CYP2D6, CYP1A4, CYP2C19, and CYP3A4 in chickens was lower than in dogs, while the activity of CYP2C9 was higher, with a K_m_ of 627.61 ± 145.61 μM in chickens [50]. Hu et al. demonstrated a similar K_m_ value of 678 ± 137 μM in Ross 708 chickens [51]. However, the K_m_ of CYP2C9 was 14.42 μM in CLM in our study, while the V_max_ was 159.4 nmol/(min·mg). Additionally, the K_m_ values of CYP1A2 and CYP3A4 were 98.1 ± 18.6 μM and 1.30 ± 0.67 μM in chickens, respectively, using PH and MDZ [51]. CYP1A1and CYP2A6 are important for biotransformation in chickens [52]. By contrast, a K_m_ of 2.1 ± 0.81 μM using MDZ in chicken was demonstrated [53]. The K_m_ for CYP1A2 and CYP3A4 in current study were 20.69 μM and 3.31 μM, respectively. The K_m_ of CYP2A6 for turkey using CAN as substrate was 33.6 ±8.52 μM [44]. In contrast, the K_m_ of CYP2A6 was 56.08 μM for chickens in our study using CAN as probe substrate.

It has been demonstrated that gene expressions of CYP3A4 and CYP3A7 in poultry was impacted by AFB1 [21]. Also, a significant decrease of CYP450 activity in rabbits was observed at an oral dose of 0.10 mg/kg/day [54]. Our results were broadly in line with above studies, which implied that AFB1 inhibited the CYP450 enzymes activity. The IC_50_ values of AFB1 for CYP2A6 and CYP3A4 were 24.11 μM and 41.37 μM in CLM, respectively, whereas there was extremely weak inhibition for CYP1A2, CYP2B6, CYP2E1, and CYP2C9 with IC_50_ values of 147.0 μM, 662.6 μM, 271.2 μM, and 526.7 μM, respectively. 

As reported, TIA inhibited CYP450 enzyme activities in rats, pigs, and goats [35,36,39]. On the other hand, another study confirmed that TIA could induce some CYP450-related catalytic activities in poultry [50]. However, our results rather excluded the possibility of TIA inhibiting the activities of enzymes in CLM. The IC_50_ values of CYP2E1 and CYP2A6 being 47.6 μM and 7.6 μM, respectively, and for CYP1A2, CYP2B6, CYP2C9, and CYP3A4 were 65.83, 101.9, 65.62, 334.2 μM, suggested weak inhibition activity in current study. A more recent study has shown that TIA was an effective inhibitor of CYP450 activities in goat and cattle microsomes [35]. This is consistent with what has been found in previous studies that TIA acted as inhibitor of porcine hepatic CYP3A and CYP2C, respectively [26]. Moreover, the CYP450 content in broiler chickens was lower in contrast with horses, pigs, cattle, rabbits, and rats [55,56]. Similarly, it was found that the rate of glucuronidation of either 1-naphthol or p-nitrophenol was in the order pigs~rabbits > horses >> cattle > broiler chicks [57].

Pharmacokinetic parameters of enrofloxacin were modified after exposure to AFB1 (750 µg/kg) in chickens [58] However, the values of T_1/2a_, T_max_ and AUC_0–∞_ of enrofloxacin were nonsignificantly increased by 17%, 26% and 17%, respectively. Also, AFB1 decreased the bioavailability and delayed the elimination of doxycycline in chickens [59]. After oral administration, the T_max_ of doxycycline in aflatoxin-exposed chickens was 2.37 h compared to 1.97 h for healthy ones. In this study, results revealed that AFB1 significantly increased the AUC_last_ of TIA when broiler chickens were given either a control or AFB1 diet for 2 weeks at a rather low contamination level (20 µg/kg feed). Hence, higher AFB1 contamination levels and/or a longer exposure period may more likely result in negative effects than a two-week exposure at a low AFB1 level of 20 μg/kg in this study [58,60]. The IC_50_ values of AFB1 and TIA were higher than 7.6 μM as reported in our in vitro results. The highest IC_50_ for TIA and AFB1 was higher than 100 μM, surpassing plasma concentrations obtained with the recommended dose for TIA and safe dose for AFB1. Meanwhile, the influence of TIA and AFB1 on the in vivo pharmacokinetics of substrates of CYP450 may not be predicted by the in vitro cocktail incubation method. Therefore, we studied the pharmacokinetic interactions between TIA and AFB1 in vivo in broilers, which indicated an interaction takes place but probably CYP450 did not play a major role in the interaction between AFB1 and TIA and other factors may be responsible for the effects observed. These findings suggested that longer exposure periods and multiple doses of AFB1 should be considered for further investigation. However, practicality and safety constraints currently limit the feasibility of such experiments. Nevertheless, a pharmacokinetic interaction between AFB1 and TIA was observed in broilers. Despite the IC_50_ of AFB1 indicating weak inhibition on CYP450 enzymes in CLM, this study provides insight that future research should focus on other factors involved in the mycotoxin-drug interaction.

The C_max_ of AFB1 in chickens reached 29.3 ng/mL at 3.2 h after oral administration at a dose of 1 mg/kg BW, indicating rapid absorption of the mycotoxin in blood. Similarly, according to literature, in rats, the T_max_ of AFB1 was about 0.15 h, with a C_max_ of 90 ± 20 ng/mL following oral administration at dose of 1 mg/kg BW [61]. Notably, the T_1/2_ was about 22.3 h in chicken following oral administration of a dose of 1 mg/kg BW. Jubert et al. reported a T_1/2α_ of 2.86 h and T_1/2β_ of 64.4 h in humans at a dose of 30 ng AFB1 [62]. In addition, in the latter study, the C_max_ and T_max_ were 0.941 ± 0.154 pg/mL and 1.02 ± 0.31 h, respectively. The AUC_0–t_ was 25.6 ± 6.3 h·pg/mL. Moreover, AFB1 shows a rapid absorption and metabolism in rats and cows compared to human [61,63]. A C_max_ of 93.42 ± 23.01 ng/mL was observed at approximately at 0.15 h in rats after a single dose of 1 mg/kg BW. The T_1/2e_ was 7.62 h in rats after oral administration, suggesting rather rapid elimination. The AUC_0–24_ was 45.59 ± 17.19 h·ng/mL, and the CL and V_d_ were 22.33 ± 9.79 L/h/kg and 226.46 ± 100.76 L/kg in rats, respectively. Guo et al. calculated the toxicokinetic parameters of AFB1 in plasma of cows after oral administration of 40 μg/kg BW, the results indicated that AFB1 was rapidly absorbed in cows with a C_max_ of 3.8 ± 0.9 ng/mL and a T_max_ of 0.58 ± 0.17 h. In addition, the T_1/2e_ and MRT were 15.52 ± 0.51 h and 11.73 ± 0.94 h, respectively, indicating that AFB1 was rather gradually eliminated in cow [64]. It was found that the AFB1 depletion in chicken liver, kidney, and muscle may take up to 10–19 days at a dose of 5 mg/kg feed [65]. In our study AFB1 had a rather short MRT of 4.7 ± 0.7 h in chickens, and the T_1/2e_ was approximately 22.3 h in broilers, which is long in contrast to rats of 7.62 h, but shorter than in humans with a T_1/2β_ of 64.4 h [62]. The primary kinetic parameters in present study were different from other animals, such as rat, cow, and human [61,62,64,66]. Overall, the difference might be due to factors such as dietary AFB1 levels, sex, age, species, duration of administration of AFB1 [65]. The limitation of this study is that only the AFB1 parent compound was analyzed, thus in future studies, the metabolites should be taken into account too, such as the highly toxic metabolite AFBO. For TIA, IC_50_ values for CYP1A2, CYP2B6, CYP2E1, CYP2C9, CYP2A6, and CYP3A4 were 65.8 μM, 101.9 μM, 47.6 μM, 65.6 μM, 7.6 μM, and 334.2 μM, respectively. The IC_50_ of AFB1 on CYP2A6, CYP3A4, CYP1A2, CYP2B6, CYP2E1 and CYP2C9 were 24.1 μM, 41.4 μM, 147.0 μM, 662.6 μM, 271.2 μM, and 526.7 μM, respectively. The C_max_ of AFB1 and TIA in broilers was 141.7 ng/mL and 44.4 ng/mL, respectively, suggesting that they were significantly lower than the IC_50_. This indicated that the concentration may not reach levels sufficient to induce an inhibitory effect. However, predicting in vivo results from in vitro data is challenging due to the complexity of CYP450 [67]. An in vivo pharmacokinetic interaction between AFB1 and TIA was observed in this study. Moreover, residues of AFB1 were detected in the liver [60] and eggs [68] at the low-level dietary exposure. Therefore, even minor changes in the pharmacokinetic characteristics of AFB1 in broilers should be carefully considered.

Pleuromutilin derivatives like TIA can inhibit CYP3A4 activity with an IC_50_ of 1.6 μM and showed strong inhibition to CYP450 enzymes [69]. In contrast, the IC_50_ values for CYP2A6 and CYP2E1 were 7.6 μM and 47.6 μM in CLM in this study, respectively. MRT (6.4 ± 3.2 h) and AUC_last_ (170 ± 103.6 h·ng/mL) of AFB1 in TIA-exposed chickens were significantly higher than TIA-unexposed chickens. Besides, the V_d_F_obs_ and CL__F_obs_ were 18.4 ± 7.7 L/kg and 9.4 ± 6.5 L/h/kg with TIA administration, respectively, while they were 14.8 ± 4.8 L/kg and 14.3 ± 6.7 L/h/kg, respectively, in broilers without TIA administration. Furthermore, Anadon et al. revealed that TIA could influence the drug metabolism rate in chickens because the pharmacokinetics of antipyrine were also altered when co-administered with TIA [70]. Of note, TIA did not have a strong inhibition on CYP1A and CYP2A in CLM in the present study. However, the metabolism of AFB1 was significantly inhibited when co-administered with TIA at a dose of 25 mg/kg in broilers as compared to AUC_last_ in the two groups. It should be noted that results obtained with microsomes cannot be used for quantitative estimations of in vivo biotransformation [71]. As TIA was reported to have a high inhibitory effect on CYP3A4 in human liver microsomes with an IC_50_ of 1.6 μM [69], data obtained in the present study exhibited that the pharmacokinetic parameters, such as AUC but not V_d_, CL, MRT, of AFB1 were altered after oral co-administration with TIA in chicken.

A pharmacokinetic interaction between AFB1 and TIA was observed, with TIA increasing the AUC of AFB1 in broilers. Considering the prominence of chicken as a global meat product, further investigations may be necessary to assess AFB1 residues in chicken tissues following exposure to TIA. Additionally, mycotoxin sorbent additives represent a type of AFB1 detoxication [72]. The combination of curcumin and cellulosic polymers have proved to be an effective approach to address poultry health issues associated with AFB1 consumption [73]. Baker yeast has been identified as partially alleviating specific toxic effects induced by AFB1 in growing chicks [74]. In an upcoming study, two types of modified general biochar will be utilized to assess their effectiveness in detoxifying AFB1.

## 4. Conclusions

The purpose of this study was to investigate the CYP450-related drug-mycotoxin interactions both in vitro and in vivo, for TIA and AFB1. The IC_50_ values of AFB1 and TIA on CYP450 in CLM suggested a weak inhibitory effect on hepatic enzymes. Nonetheless, the pharmacokinetic profiles of AFB1 and TIA were somewhat altered with co-exposure, which indicated that CYP450 did not play a major role in the interaction between AFB1 and TIA. It is crucial to delve further into exploring additional factors involved in the interaction. Furthermore, it is essential to manage the use of antibiotics and feed contaminated with mycotoxins in poultry production to minimize potential risks to both poultry health and food safety.

## 5. Materials and Methods

### 5.1. Chemicals and Reagents

TIA was obtained from Fisher Scientific (Fair Lawn, NJ, USA). AFB1 was purchased from Fermentek (Jerusalem, Israel). MDZ was obtained from Roche (Mannheim, Germany). PH was obtained from Sigma Aldrich (St. Louis, MO, USA). 4-acetaminophen (ACE), 1-hydroxymidazolam (OH-MDZ), CLN, 6-hydroxychlorzoxazone (OH-CLN), BP, hydroxybupropion (OH-BP), TBD, 4-hydroxytolbutamide (OH-TBD), CAN, 7-hydroxycoumarin (OH-CAN), and β-nicotinamide adenine dinucleotide 2′-phosphate reduced tetrasodium salt hydrate (NADPH.4Na) were purchased from Fisher Scientific (Fair Lawn, NJ, USA). Chicken liver microsomes (CLM) were purchased from Primacyt (Schwerin, Germany). LC-MS grade acetonitrile (ACN) and methanol were purchased from Biosolve B.V. (Valkenswaard, The Netherlands). Formic acid (FA) was purchased from Merck (Darmstadt, Germany). Water was from an ultrapure water system (Sartorius, Goettingen, Germany). A phosphate buffer (PBS) was prepared using a tablet purchased from Fisher Scientific (Fair Lawn, NJ, USA). The 0.1 M PBS (pH = 7.4) was obtained by dissolving the tablet in 200 mL ultrapure water. The commercial antibiotic product Vetmulin^®^ (450 mg tiamulin hydrogen fumarate/g) was purchased from Huvepharma (Sofia, Bulgaria).

### 5.2. Microsomal Incubation Procedure

A cocktail-approach for elucidating the metabolites profiling and CYP450 inhibition was conducted as in Lin et al. with minor changes [75]. The incubation mixture contained 50 μL of 0.2 mg/mL of CLM, 10 μL of 0.1 M phosphate buffer (pH 7.4), 40 μL of 1 mM NADPH and six probe substrates with a total volume of 200 μL. The specific probe substrates used for each CYP450 isoenzyme in the incubation mixture were as follows: PH (CYP1A2), BP (CYP2B6), CLN (CYP2E1), TBD (CYP2C9), CAN (CYP2A6), and MDZ (CYP3A4). Probes substrates were dissolved in methanol as stock solution (1 mg/mL). All of them were diluted to the final concentrations (see Appendix A) with incubation buffer and the percentage of organic solvent in the final incubation volume was below 1% (*v*/*v*) [76]. Briefly, the mixture was preincubated in a thermoshaker (Biosan, Geraardsbergen, Belgium) for 5 min at 37 °C prior to the initiation of the reaction with NADPH.4Na. After incubation, the mixture was terminated by 400 μL ice-cold acetonitrile. Each sample was centrifuged (5810R, Eppendorf, Germany) at 14,000 rpm for 10 min. A volume of 5 µL of supernatant was analysed using UHPLC-MS/MS.

Enzyme kinetic studies were conducted for the six CYP450-related probes reactions. For each probe, the maximum rate of the V_max_ and K_m_ were obtained using GraphPad Prism v.8.0.2 software via Michaelis-Menten Kinetic model (La Jolla, CA, USA). The incubation times of 5, 10, 15, 30, 45 and 60 min were tested individually. A series of protein concentrations (0.05, 0.1, 0.2, 0.4, 0.6, 0.8 and 1 mg/mL) of CLM in the incubation mixture were conducted. Triplicate analyses were performed for each incubation system. The optimal values for the incubation time and CLM protein concentration were found to be 15 min for PH, BP, and CAN, 20 min for CLN, 5 min for MDZ, and TBD, respectively, and 0.4 mg/mL protein concentration for PH, BP, MDZ, and CLN, 0.2 mg/mL for TBD, and CAN, respectively.

Inhibition assays of TIA and AFB1 on CYP450 isoforms in CLM were performed with a fixed concentration of 11.2 μM for the model probes substrate CYP1A2, 4.2 μM for CYP2B6, 35.4 μM for CYP2E1, 14.7 μM for CYP2C9, 136.8 μM for CYP2A6, and 2.5 μM for 3A4, respectively. For the inhibition assays, the final concentration of AFB1 and TIA in the incubation mixture ranged from 0–25 μM and 0–40 μM, respectively.

### 5.3. Chromatographic and Mass Spectrometric Conditions

Liquid chromatography was performed on an ACQUITY UHPLC system (Waters, Milford, MA, USA) coupled with a XEVO TQS triple quadrupole tandem mass spectrometer (Waters, Milford, MA, USA). The mobile phase comprised solution A (0.1% FA in water) and solution B (0.1% FA in methanol). The gradient elution program was performed as follows: 0–1.5 min (5–15% B); 1.5–3.5 min (15–95% B); 3.5–7.5 min (95% B), 7.5–9.5 min (95–5% B), 9.5–10.0 min (5% B). The injection volume was 5 μL and the flow rate was 0.3 mL/min. An Acquity HSS T3 C18 column (100 mm × 2.1 mm, 1.7 μm) was used to perform the analysis at a temperature of 40 °C.

The mass spectrometer contained an electrospray ionization (ESI) source. The ESI source was operated in both positive and negative ionization mode with capillary voltage, 3.0 kV; source temperature, 120 °C; desolvation temperature, 500 °C; desolvation gas flow rate, 100 L/h; and cone gas flow rate, 150 L/h. The MS conditions for all the analytes are shown in the Table 6.

### 5.4. Animal Experiments Design

#### 5.4.1. Pharmacokinetic Study of AFB1 with or without TIA

Eight broiler chickens (4-week-old, 4 male/4 female, Ross 308) weighing 1.8–2.5 kg were provided by the College of Veterinary Medicine (Beijing, China), and authorization was approved by the Animal Care and Use Committee of China Agricultural University on 12 April 2022 (39905-23-G-001). The smaller number of chickens was used due to the limited availability of AFB1. A commercial broiler grower diet (511, CP feed, Tianjin, China) was used, that contained no antibiotics and AFB1 was not found in the feed. Animals were handled in accordance with the guideline for Care and Use of laboratory Animals. Chickens were selected randomly and numbered according to random numbers generated in Excel. They were housed individually under controlled conditions (22 °C, 18:6 h light/dark cycle) with free access to feed and water throughout the experiment. All broiler chickens went through 8 h fasting before each oral gavage dosing. The treatment group was given TIA at a dose of 25 mg/kg BW by gavage for 4 consecutive days. The control group was given the same volume of saline as the TIA solution for 4 days. After TIA administration at day 4, two groups were orally administered AFB1 at a dose of 1 mg/kg BW at 30 min after TIA administration. The chickens received feed again after oral administration of AFB1. At 0.25, 0.5, 1, 2, 3, 4, 6, 8, 12 and 24 h after oral administration (p.a.), blood was obtained (0.5 mL) via direct venepuncture of the leg vein and put into heparinized tubes. This was followed with centrifugation (TGL16M, Xiangli, China) (4000 rpm, 10 min) to obtain plasma, which was stored at −20 °C until analysis for AFB1.

#### 5.4.2. Pharmacokinetic Study of TIA with or without AFB1

A commercial broiler grower diet (Farmer Mash, Aveve, Belgium) was fed to all chickens. Screening of this feed was executed by a multi-mycotoxin LC-MS/MS method, indicating it was free of AFB1. Chickens were fed either the grower diet without AFB1, or a diet experimentally contaminated with 20 µg/kg AFB1 [45]. To produce the grower diet experimentally contaminated with AFB1, first, 100 µL of AFB1 working solution (200 µg/mL in edible grade ethanol) was mixed with 1 g of feed for every batch. In total, 40 batches of contaminated feed were prepared. Then, this premix was blended with 1 kg of feed for 30 min to ensure a homogenous distribution of AFB1, with the results confirmed by UHPLC-MS/MS that the samples were contaminated with AFB1 in a range from 9.3 to 25.4 μg/kg (see Appendix A). Moreover, daily feed consumption was recorded.

Eighteen Ross 308 chickens (2-week-old, male) weighing 0.5–1.1 kg were provided by a commercial farm (Ghent, Belgium) and were randomly divided in 2 groups of 9 birds each. Chickens were selected randomly and numbered according to random numbers generated in Excel. The animal experiment was approved by the Ethical Committee of the Faculty of Veterinary Medicine and of Bioscience Engineering of Ghent University on 23 November 2022 (EC 2022-068). They were housed individually under controlled conditions (22 °C, 18:6 h light/dark cycle) with free access to feed and water throughout the experiment. All broiler chickens went through 8 h fasting before the bolus administration of TIA, and were fed again 4 h p.a. While the treatment group was fed with an AFB1-contaminated diet for 2 weeks, the control group was given an AFB1-free grower diet. Each animal was orally administered with a single bolus of TIA at a dose of 25 mg/kg BW after two consecutive weeks of AFB1 exposure. Furthermore, chickens were given TIA by drinking water (25 mg/kg BW) for four consecutive days after the oral bolus administration of TIA. New medicated drinking water was prepared daily. To calibrate the dose, the volume of the drinking water was noted. After bolus dosing, blood was collected into heparinized tubes by direct venepuncture of the leg vein at 0.25, 0.5, 1, 2, 3, 4, 6, 8, 12, 24, 36, 48, 60, 72, 84, 96, 108, 120, 132, 144, 156, 168, and 180 h p.a. and put into heparinized tubes. Then, the blood samples were centrifuged (5810R, Eppendorf, Germany) at 4000 rpm for 10 min to obtain the plasma, which was stored at −20 °C until analysis for TIA. After the last blood sampling, birds were euthanized and liver samples were collected for histopathological analysis.

### 5.5. Plasma Sample Preparation and UHPLC-MS/MS Analysis

A volume of 100 µL of chicken plasma was added to 150 µL of ACN in a 1.5 mL centrifuge tube. The samples were vortexed for 5 min and then centrifuged (5810R, Eppendorf, Germany) at 14,000 rpm for 10 min. Analysis was performed by injecting the supernatant (5 µL) into the UHPLC-MS/MS instrument. The instrumental condition of TIA and AFB1 was showed in Appendix A.

### 5.6. Statistical Analysis

All statistical analyses were performed using Graph Pad Prism 8.0.2 software. The results obtained from the data analysis were expressed as mean ± standard deviation (SD). The pharmacokinetic parameters were calculated by WinNonlin software (v.8.3.4). T_1/2λz_, elimination half-life, $T_{1/2\lambda_{Z}}=\ln2/\lambda_{Z}$. (The terminal elimination rate constant (λ_Z_), obtained by taking a semi-logarithmic linear regression from the elimination phase). T_max_, time to reach peak plasma concentration after administration. C_max_, maximum plasma concentration after administration.

AUC_last_, area under the concentration-time curve from 0 to the last time point, $AUC\int_{t1}^{t2}=\Delta t\times \frac{\left(C1+C2\right)}{2}$. AUC_INF_obs_, area under the concentration-time curve from 0 to infinity, $AUC_{INF_{obs}}=AUC_{last}+\frac{C_{last}}{\lambda_{z}}$. V_Z_F_obs_, apparent volume of distribution, $V_{Z\_F\_obs}=\frac{Dose}{\lambda_{z}\times V_{z\_F\_obs}}$. Cl__F_obs_, apparent body clearance, $Cl_{\_F\_obs}=\frac{Dose}{AUC_{INF\_obs}}$. MRT_last_, mean residence time, $MRT_{last}=\int_{0}^{t}C\times tdt$. The difference between pharmacokinetic parameters was analysed by SPSS software version 26. A *p*-value < 0.05 was regarded as significant.

## Figures and Tables

**Figure 1 toxins-16-00160-f001:**
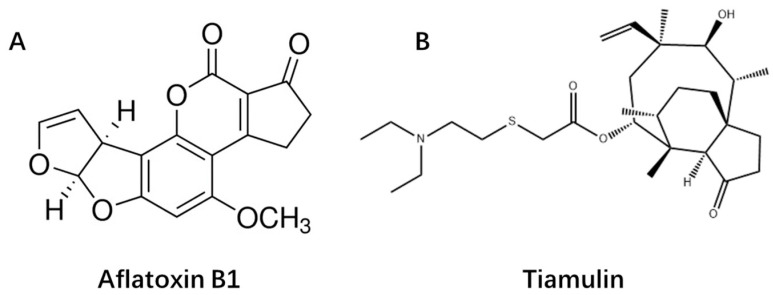
The chemical structure of aflatoxin B1 (**A**) and Tiamulin (**B**).

**Figure 2 toxins-16-00160-f002:**
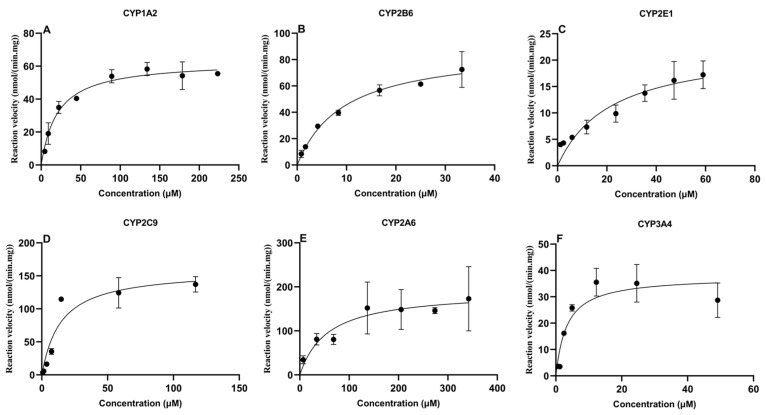
Michaelis-Menten plots of six CYP450 probes in chicken liver microsomes (mean ± SD, *n* = 3): (**A**) phenacetin; (**B**) bupropion; (**C**) chlorzoxazone; (**D**) tolbutamide; (**E**) coumarin; and (**F**) midazolam.

**Figure 3 toxins-16-00160-f003:**
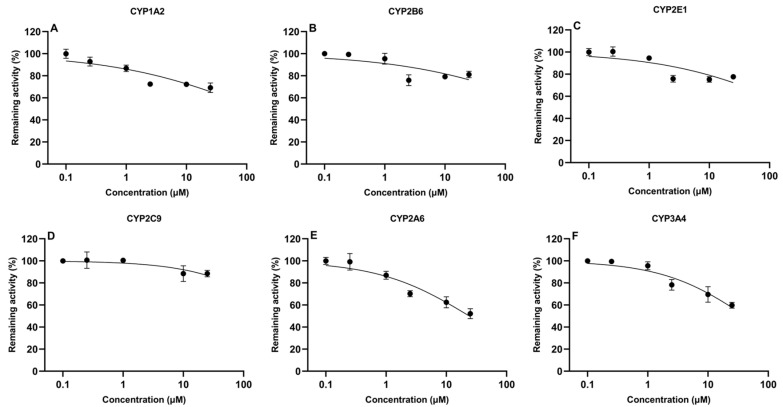
In vitro inhibitory effect of aflatoxin B1 on cytochrome CYP450 isoforms in chicken liver microsomes (mean ± SD, *n* = 3): (**A**) CYP1A2; (**B**) CYP2B6; (**C**) CYP2E1; (**D**) CYP2C9; (**E**) CYP2A6; and (**F**) CYP3A4.

**Figure 4 toxins-16-00160-f004:**
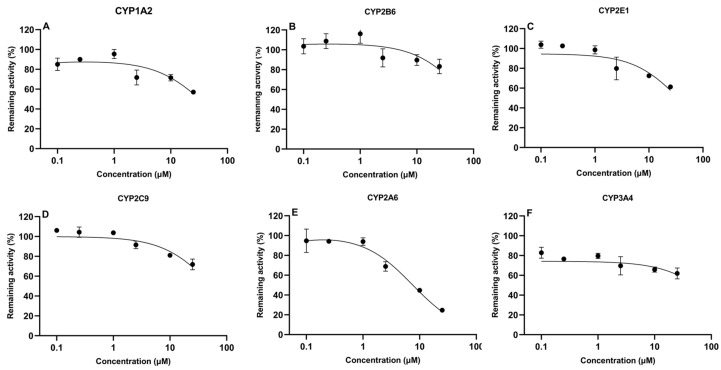
In vitro inhibitory effect of tiamulin on cytochrome CYP450 isoforms in chicken liver microsomes (mean ± SD, *n* = 3): (**A**) CYP1A2; (**B**) CYP2B6; (**C**) CYP2E1; (**D**) CYP2C9; (**E**) CYP2A6; and (**F**) CYP3A4.

**Figure 5 toxins-16-00160-f005:**
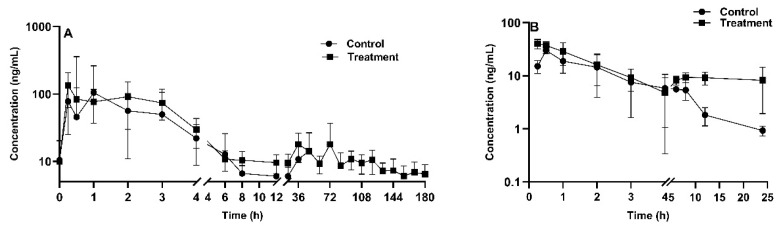
Tiamulin plasma concentration-time profile after oral administration in chickens (*n* = 8 in the treatment group, *n* = 9 in the control group) with (treatment) or without (control) aflatoxin B1 contaminated diet (20 µg/kg feed for 2 weeks) (**A**); aflatoxin B1 plasma concentration-time profile after oral administration in chickens (*n* = 4 in each group) with (treatment) or without (control) tiamulin oral gavage administration (25 mg/kg BW for 4 days) (**B**). The mean ± SD is shown.

**Table 1 toxins-16-00160-t001:** The intra-day and inter-day accuracy and precision of six probe substrates chicken liver microsomes.

Analytes	Concentration	Intra-Day (*n* = 5)		Inter-Day (*n* = 15)	
	(ng/mL)	Accuracy (% Mean ± SD)	Precision (% CV)	Accuracy (% Mean ± SD)	Precision (% CV)
Acetaminophen	5	92.5 ± 8.8	9.5	92.8 ± 12.0	12.9
	20	101.2 ± 8.8	8.7	101.8 ± 7.8	7.6
	100	105.4 ± 0.5	0.4	102.2 ± 6.4	6.3
OH-Bupropion	5	96.6 ± 13.0	13.5	96.2 ± 12.3	12.8
	20	99.2 ± 10.0	10.1	103.9 ± 11.1	10.7
	100	104.8 ± 1.6	1.5	101.2 ± 6.8	6.8
OH-Chlorzoxazone	5	94.0 ± 8.5	12.7	99.7 ± 12.9	12.9
	20	98.3 ± 8.5	8.7	94.6 ± 7.1	7.5
	100	99.8 ± 9.5	9.5	94.7 ± 8.6	9.1
OH-Tolbutamide	5	102.8 ± 1.0	1.0	97.4 ± 11.6	11.9
	20	97.4 ± 8.1	8.3	96.1 ± 7.7	8.1
	100	96.9 ± 2.5	2.6	96.0 ± 7.4	7.7
OH-Coumarin	5	98.2± 11.3	11.5	95.2 ± 9.5	9.9
	20	108.5 ± 6.9	6.4	103.8 ± 11.8	11.0
	100	108.6 ± 7.9	7.3	103.3 ± 10.7	10.4
OH-Midazolam	5	95.6 ± 12.6	13.2	90.4 ± 12.5	13.8
	20	99.7 ± 5.8	5.8	102.0 ± 6.6	6.5
	100	97.4 ± 7.3	7.5	98.2 ± 5.7	5.8

**Table 2 toxins-16-00160-t002:** The intra-day and inter-day accuracy and precision of tiamulin and aflatoxin B1 in chicken plasma.

Analytes	Concentration	Intra-Day (*n* = 5)		Inter-Day (*n* = 15)	
	(ng/mL)	Accuracy (% Mean ± SD)	Precision (% CV)	Accuracy (% Mean ± SD)	Precision (% CV)
Tiamulin	10	96.3± 5.0	5.2	96.3 ± 8.6	8.9
	250	96.9 ± 5.3	5.4	96.9 ± 5.9	6.1
	5000	95.6 ± 2.8	3.7	97.4 ± 4.7	4.8
Aflatoxin B1	0.5	95.4 ± 3.4	3.6	95.4 ± 4.1	4.3
	5	99.9 ± 2.5	2.5	99.9 ± 4.8	4.8
	20	97.3 ± 1.3	1.3	97.3 ± 2.6	2.7

**Table 3 toxins-16-00160-t003:** Incubation conditions for six probe substrates for the in vitro CYP450 inhibition study.

Analytes	Protein Concentration (mg/mL)	Substrate Concentration (μM)	Incubation Time (min)	K_m_ (μM)	V_max_ (nmol/(min·mg))
Phenacetin	0.4	11.2	15	20.7	63.1
Bupropion	0.4	4.2	15	9.4	88.9
Chlorzoxazone	0.4	35.4	20	21.8	22.7
Tolbutamide	0.2	14.7	5	14.4	159.4
Coumarin	0.2	136.8	15	56.1	190.8
Midazolam	0.4	2.5	5	3.3	37.7

K_m_, Michaelis-Menten constant; V_max_, maximal velocity.

**Table 4 toxins-16-00160-t004:** Pharmacokinetic parameters of tiamulin (TIA) in chickens pretreated with or without aflatoxin B1 contaminated diet (20 µg/kg feed for 2 weeks).

Parameters	TIA	
	Treatment ^1^	Control ^2^
T_1/2λz_ (h)	6.3 ± 2.2	7.0 ± 2.9
T_max_ (h)	1.0 ± 1.1	0.4 ± 0.2
C_max_ (ng/mL)	141.7 ± 65.4	89.9 ± 45.9
AUC_last_ (h·ng/mL)	499.8 ± 155.5	294.5 ± 63.3 **
AUC_INF_obs_ (h·ng/mL)	585.8 ± 173.9	357.3 ± 69.6 **
V_d_F_obs_ (L/kg)	407.1 ± 142.5	715.1 ± 228.45 **
CL__F_obs_ (L/h/kg)	45.8 ± 12.3	72.3 ± 13.8 ***
MRT_last_ (h)	6.2 ± 1.8	6.8 ± 1.4

^1^ Tiamulin pharmacokinetic parameters in chickens pretreated with aflatoxin B1 contaminated diet (20 µg/kg feed for 2 weeks) (*n* = 8); ^2^ Tiamulin pharmacokinetic parameters in chickens that were not pretreated with aflatoxin B1 contaminated diet (control feed for 2 weeks) (*n* = 9). The AUC_last_ of tiamulin was calculated until last sampling point of 180 h. ** Significantly different from control, *p* < 0.01. *** Significantly different from control, *p* < 0.001. The mean ± SD is shown. T_1/2λz_: elimination half-life; T_max_: time to reach peak plasma concentration; C_max_: plasma peak concentration; AUC_last_: area under the concentration-time curve from 0 to the last time point; AUC_INF_obs_: area under the concentration-time curve from 0 to infinity; V_d_F_obs_: apparent volume of distribution; CL__F_obs_: apparent body clearance; MRT_last_: mean residence time.

**Table 5 toxins-16-00160-t005:** Pharmacokinetic parameters of aflatoxin B1 (AFB1) in chickens pretreated with or without tiamulin (25 mg/kg of BW for 4 days).

Parameters	AFB1	
	Treatment ^1^	Control ^2^
T_1/2λz_ (h)	1.4 ± 0.5	1.13 ± 0.85
T_max_ (h)	0.3 ± 0.1	1 ± 0.6
C_max_ (ng/mL)	44.4 ± 4.7	29.3 ± 3.2 **
AUC_last_ (h·ng/mL)	170 ± 103.6	87.5 ± 39.6 **
AUC_INF_obs_ (h·ng/mL)	265.8 ± 222.9	94.2 ± 40.4 **
V_d_F_obs_ (L/kg)	18.4 ± 7.7	14.8 ± 4.8
CL__F_obs_ (L/h/kg)	9.4 ± 6.5	14.3 ± 6.7
MRT_last_ (h)	6.4 ± 3.2	4.7 ± 0.7

^1^ Aflatoxin B1 pharmacokinetic parameters in chickens pretreated with tiamulin (25 mg/kg BW for 4 days) (*n* = 4); ^2^ Aflatoxin B1 pharmacokinetic parameters in chickens that were not pretreated with tiamulin (25 mg/kg BW saline for 4 days) (*n* = 4). The AUC_last_ of aflatoxin B1 was calculated until last sampling point of 24 h. ** Significantly different from control, *p* < 0.01. The mean ± SD is shown. T_1/2λz_: elimination half-life; T_max_: time to reach peak plasma concentration; C_max_: plasma peak concentration; AUC_last_: area under the concentration-time curve from 0 to the last time point; AUC_INF_obs_: area under the concentration-time curve from 0 to infinity; V_d_F_obs_: apparent volume of distribution; CL__F_obs_: apparent body clearance; MRT_last_: mean residence time.

**Table 6 toxins-16-00160-t006:** Mass spectrometry parameters for six probes substrates, tiamulin, and aflatoxin B1.

Analytes	Ionization Mode	Precursor Ion (*m*/*z*)	Product Ion (*m*/*z*)	Cone (V)	Collision (eV)
Acetaminophen	ESI+	152.1	110	40	10
			134	40	8
OH-Bupropion	ESI+	256	167	25	20
			184	25	20
OH-Chlorzoxazone	ESI-	183.8	148	20	16
			120	20	18
OH-Tolbutamide	ESI+	285	104	35	22
			186	35	17
OH-Coumarin	ESI+	161.9	106	30	19
			134	30	20
OH-Midazolam	ESI+	342	203	25	22
			289	25	22
Tiamulin	ESI+	494.4	119	35	41
			192	35	44
Aflatoxin B1	ESI+	313	241	/	32
			285	/	20

## Data Availability

The data can be made available if required.

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
