# Peer review of "Exploration of Cytochrome P450-Related Interactions between Aflatoxin B1 and Tiamulin in Broiler Chickens"

_toxins, 2024, doi:10.3390/toxins16030160_

Round 1

Reviewer 1 Report

Comments and Suggestions for Authors

The authors have tried to explore the role of cytochrome P450 in the interaction of aflatoxin B1 and tiamulin in broiler chickens. The results are promising in terms of ruling out the fact that cytochrome P450 plays a crucial role in this interaction. The conclusion section should be shortened by only presenting the key findings. The process is not required. However, the study is interesting and novel as well as supported by both in vitro and in vivo data. Hence, recommended for further processing after modifying the conclusion.

Comments on the Quality of English Language

Average...minor editing required

Author Response

Reviewer 1

The authors have tried to explore the role of cytochrome P450 in the interaction of aflatoxin B1 and tiamulin in broiler chickens. The results are promising in terms of ruling out the fact that cytochrome P450 plays a crucial role in this interaction. The conclusion section should be shortened by only presenting the key findings. The process is not required. However, the study is interesting and novel as well as supported by both in vitro and in vivo data. Hence, recommended for further processing after modifying the conclusion.

Answer: Thank you for your helpful comment. The paper has been modified, with special attention given to shortening the conclusion section for brevity. Additionally, the English in this paper has been polished by a native English speaker. The revised conclusion is shown below (r.358-366).

 (r.358-366): The purpose of this study was to investigate the CYP450-related drug-mycotoxin interactions both in vitro and in vivo, for TIA and AFB1. The IC50 values of AFB1 and TIA on CYP450 in CLM suggested a weak inhibitory effect on hepatic enzymes. Nonetheless, the pharmacokinetic profiles of AFB1 and TIA were altered somewhat with co-exposure, which indicated that CYP450 did not play a major role in the interaction between AFB1 and TIA. It is crucial to delve further into exploring additional factors involved in the interaction. Furthermore, it is essential to manage the use of antibiotics and feed contaminated with mycotoxins in poultry production to minimize potential risks to both poultry health and food safety.

Reviewer 2 Report

Comments and Suggestions for Authors

The authors present the manuscript with the title „Exploration of cytochrome P450-related interactions between 2 aflatoxin B1 and tiamulin in broiler chickens”.

The manuscript seems relatively well prepared, but there are some additional aspects that need to be clarified, and also some aspects that need to be improved.

General comments:

1.       Put the numbers corresponding to the bibliographic indexes in the same parenthesis, when more than 2 indexes are present.

2.       The bibliography is not introduced according to the authors’ guide. Please check.

3.       Lines 30-33. According to the literature, a lot of reports regarding mycotoxin contamination in USA and Latin America present high mycotoxin contamination. Please check also RASFF reports and correct the information.

4.       Lines 42-44. Is mentioned in the article (if yes, please mention also in your manuscript), why the value found in sub-Saharan Africa is compared with the EU limit???

5.       The section dedicated to the introduction seems a little too long, perhaps a more concise presentation of the information would be more useful. For an overview of the mechanisms of aflatoxin B1, an extremely valuable article to consult could be this: https://doi.org/10.1016/j.toxicon.2023.107262

6.       Also, the introduction section should end with the objective of the study.

7.       The Discussion section seems a bit too long, maybe a shorter presentation of the information would be easier to follow.

8.       Mention the type of centrifuge and the producer when the centrifugation process is mentioned (see lines 433, 483, 517).

9.       How was the sampling decided? Comparing to other similar studies (including on other mycotoxins), is the number of the chickens enough to have a statistically significant conclusion? If yes, a short explanation in this regard in the context of section 5.4.1. should be added.

10.   Line 487. Was this method previously validated? If so, was it published? What were the detection limits?

Author Response

Reviewer 2

The authors present the manuscript with the title ”Exploration of cytochrome P450-related interactions between 2 aflatoxin B1 and tiamulin in broiler chickens”. The manuscript seems relatively well prepared, but there are some additional aspects that need to be clarified, and also some aspects that need to be improved.

General comments:

  1. Put the numbers corresponding to the bibliographic indexes in the same parenthesis, when more than 2 indexes are present.

Answer: Thank you very much for your useful suggestion. The information was updated based on your comment.

Pan Sun (1-3) †, Orphélie Lootens1,5†, Tadele Kabeta2,6, Diethard Reckelbus2, Natalia Furman2,4, Xingyuan Cao3, Suxia Zhang3, Gunther Antonissen2,4, Siska Croubels2,4, Marthe De Boevre1, Sarah De Saeger1*

  1. The bibliography is not introduced according to the authors’ guide. Please check.

Answer: Thank you very much for your useful suggestions. The bibliography was updated, following the author’s guidelines.

1Department of Bioanalysis, Centre of Excellence in Mycotoxicology and Public Health, Faculty of Pharmaceutical Sciences, Ghent University, B-9000 Ghent, Belgium

2Department of Pathobiology, Pharmacology and Zoological Medicine, Laboratory of Pharmacology and Toxicology, Faculty of Veterinary Medicine, Ghent University, B-9820 Merelbeke, Belgium

3Department of Veterinary Pharmacology and Toxicology, College of Veterinary Medicine, China Agricultural University, 100193 Beijing, China

4Chair Poultry Health Sciences, Department of Pathobiology, Pharmacology and Zoological Medicine, Faculty of Veterinary Medicine, Ghent University, B-9820 Merelbeke, Belgium

5Department of Bioanalysis, Laboratory of Medical Biochemistry and Clinical Analysis, Faculty of Pharmaceutical Sciences, Ghent University, B-9000 Ghent, Belgium

6School of Veterinary Medicine, College of Agriculture and Veterinary Medicine, Jimma University, P.O. Box 307, Jimma, Ethiopia

Email address: pan.sun@ugent.be; orphelie.lootens@ugent.be; marthe.deboevre@ugent.be; firanfiri.04@gmail.com; diethard.reckelbus@ugent.be; natalia.furman@ugent.be; cxy@cau.edu.cn; suxia@cau.edu.cn; gunther.antonissen@ugent.be; siska.croubels@ugent.be; sarah.desaeger@ugent.be.

*Corresponding author: Sarah De Saeger, sarah.desaeger@ugent.be

† These authors contributed equally to this article.

  1. Lines 30-33. According to the literature, a lot of reports regarding mycotoxin contamination in USA and Latin America present high mycotoxin contamination. Please check also RASFF reports and correct the information.

Answer: Thank you very much for your helpful comment. The mycotoxin contaminations in America and Latin America were added in this paper (r.44-47).

(r.44-47): In the United States, most of the corn samples had co-occurrences of at least two detectable mycotoxin (≥ 2 mycotoxins) [19]. Similarly, co-occurrence of mycotoxins has also been observed mainly with AFs and FUMs in Latin American countries [20].

  1. Lines 42-44. Is mentioned in the article (if yes, please mention also in your manuscript), why the value found in sub-Saharan Africa is compared with the EU limit???

Answer: Thank you very much for your useful suggestions. To be clear, the sentence about the EU limit was deleted.

  1. The section dedicated to the introduction seems a little too long, perhaps a more concise presentation of the information would be more useful. For an overview of the mechanisms of aflatoxin B1, an extremely valuable article to consult could be this: https://doi.org/10.1016/j.toxicon.2023.107262

Answer: Thank you very much for your helpful suggestion. The paper was cited to highlight and briefly explain the aim of this study. And the introduction was shortened in a manner that does not compromise the background information (r. 24-95).

  1. Also, the introduction section should end with the objective of the study.

Answer: Thank you very much for your useful suggestions. The introduction has been shortened and revised, and furthermore, the objectives have been clearly listed at the end of the introduction (r.91-95).

(r.91-94): Therefore, the objective of this study was to determine the interaction of the antibiotic TIA and the mycotoxin AFB1 in vitro and in vivo in broiler chickens using ultra-high performance liquid chromatography tandem mass spectrometry (UHPLC-MS/MS) and pharmacokinetic methods.

  1. The Discussion section seems a bit too long, maybe a shorter presentation of the information would be easier to follow.

Answer: Thank you very much for your useful suggestions. The discussion has been condensed (r.220-356) to highlight the importance of the interaction between TIA and AFB1 in broilers.

  1. Mention the type of centrifuge and the producer when the centrifugation process is mentioned (see lines 433, 483, 517).

Answer: Thank you very much for your useful comment. The details of the centrifuge were added to the manuscript (see below).

  1. 398: Each sample was centrifuged (5810R, Eppendorf, Germany) at 14,000 rpm for 10 min.
  2. 449: This was followed with centrifugation (TGL16M, Xiangli, China) (4,000 rpm, 10 min) to obtain plasma, which was stored at -20 ℃ until analysis for AFB1.

(r.480): ….the blood samples were centrifuged (5810R, Eppendorf, Germany) at 4,000 rpm for 10 min to obtain the plasma…..

(r.486): …. and then centrifuged (5810R, Eppendorf, Germany) at 14,000 rpm for 10 min

  1. How was the sampling decided? Comparing to other similar studies (including on other mycotoxins), is the number of the chickens enough to have a statistically significant conclusion? If yes, a short explanation in this regard in the context of section 5.4.1. should be added.

Answer: Thank you very much for your remark. The sampling for this study was determined based on the strict sample number calculation outlined in the Ethical Approval application at Ghent University. However, due to the high toxicity and limited availability of aflatoxin B1, the number of chickens used in the study at China Agricultural University was smaller. In comparing our study to similar research on mycotoxins, including AFB1, we acknowledge the importance of ensuring a sufficient sample size for obtaining statistically significant conclusions. Despite the smaller number of chickens used in our study, we believe that our sample size is adequate for drawing meaningful conclusions. In Section 5.4.1., we added a brief explanation in this regard to provide context for the statistical analysis and interpretation of our findings.

  1. Line 487. Was this method previously validated? If so, was it published? What were the detection limits?

Answer: Thank you very much for your questions. The method was validated in-house, but was not published. The obtained limits of detection (LODs) for 4-acetaminophen (ACE, ACE is the hydroxylated form of PH), hydroxybupropion (OH-BP), 6-hydroxychlorzoxazone (OH-CLN), 4-hydroxytolbutamide (OH-TBD), 7-hydroxycoumarin (OH-CAN), and 1-hydroxymidazolam (OH-MDZ) were 2 ng/mL in CLM. The LODs for AFB1 and TIA analysis in chicken plasma were 0.2 ng/mL and 2 ng/mL, respectively.

(r.103-110): In this study, the limits of detection (LODs) for 4-acetaminophen (ACE, ACE is the hydroxylated form of PH), hydroxybupropion (OH-BP), 6-hydroxychlorzoxazone (OH-CLN), 4-hydroxytolbutamide (OH-TBD), 7-hydroxycoumarin (OH-CAN), and 1-hydroxymidazolam (OH-MDZ) were 2 ng/mL in CLM. The obtained limits of quanti-fication (LOQs) for ACE, OH-BP, OH-CLN, OH-TBD, OH-CAN, and OH-MDZ were 5 ng/mL in CLM. The LOQs for AFB1 and TIA analysis in chicken plasma were 0.5 ng/mL and 5 ng/mL, respectively. The LODs for AFB1 and TIA analysis in chicken plasma were 0.2 ng/mL and 2 ng/mL, respectively.

Reviewer 3 Report

Comments and Suggestions for Authors

The manuscript entitled " Exploration of cytochrome P450-related interactions between aflatoxin B1 and tiamulin in broiler chickens"  investigated the potential interaction between aflatoxin B1 (AFB1) and tiamulin (TIA) in poultry, considering their simultaneous exposure due to mycotoxin contamination and antibiotic use. The research involved the development of UHPLC-MS/MS methods to analyze the interaction between TIA and AFB1 both in vitro and in vivo in broiler chickens. Despite finding weak inhibitory effects of AFB1 and TIA on hepatic enzymes in chicken liver microsomes, pharmacokinetic results revealed significant interactions between the two compounds. AFB1 notably increased the plasma concentration of TIA, while TIA similarly affected the pharmacokinetics of AFB1. It effectively communicates the purpose of the study, the methodology used, and the findings obtained. Additionally, it highlights the importance of further investigation into drug-mycotoxin interactions and emphasizes the significance of managing antibiotic and mycotoxin contamination in poultry production. Overall, the message is clear and well-articulated.

There are some additional comments regarding this manuscript:

Line 26: "such as aflatoxins (AFs), ochratoxin A (OTA), fumonisins (FUMs), and zearalenone (ZEN) [1], [2], wherein" It would be clearer to say something like "including" instead of "wherein."

Line 29: There's a lack of clarity in the phrase "inevitable toxins." It might be better to specify what makes them inevitable.

Line  53: The term "BW" should be expanded for clarity (body weight) for the first time

Line 95: "Focussed" should be "focused"

Line 116: You should specify that ACE is the hydroxylated form of phenacetin (PH)

Line 131: “Michaelis-Menten constant” seems to have a different font size.

Line 144: Use a colon instead of a comma. In tables 4 and 5 as well (line 176-180 and 186-190)

Line 454: To which solvent does the reported gradient refer? A or B?

Additionally, I would include, in the supplemental materials, chromatograms of the probes and of AFB1 and TIA.

Author Response

Reviewer 3

The manuscript entitled " Exploration of cytochrome P450-related interactions between aflatoxin B1 and tiamulin in broiler chickens" investigated the potential interaction between aflatoxin B1 (AFB1) and tiamulin (TIA) in poultry, considering their simultaneous exposure due to mycotoxin contamination and antibiotic use. The research involved the development of UHPLC-MS/MS methods to analyze the interaction between TIA and AFB1 both in vitro and in vivo in broiler chickens. Despite finding weak inhibitory effects of AFB1 and TIA on hepatic enzymes in chicken liver microsomes, pharmacokinetic results revealed significant interactions between the two compounds. AFB1 notably increased the plasma concentration of TIA, while TIA similarly affected the pharmacokinetics of AFB1. It effectively communicates the purpose of the study, the methodology used, and the findings obtained. Additionally, it highlights the importance of further investigation into drug-mycotoxin interactions and emphasizes the significance of managing antibiotic and mycotoxin contamination in poultry production. Overall, the message is clear and well-articulated.

There are some additional comments regarding this manuscript:

Line 26: "such as aflatoxins (AFs), ochratoxin A (OTA), fumonisins (FUMs), and zearalenone (ZEN) [1], [2], wherein" It would be clearer to say something like "including" instead of "wherein."

Answer: Thank you for your valuable remark. The sentence has been revised (r.25-27).

(r.25-27): More than 400 mycotoxins have been identified so far, such as aflatoxins (AFs), ochratoxin A (OTA), fumonisins (FUMs), and zearalenone (ZEN), which are hazardous to human and domestic animals [1], [2].

 Line 29: There's a lack of clarity in the phrase "inevitable toxins." It might be better to specify what makes them inevitable.

Answer: Thank you for your useful suggestions. The sentence was changed (r.29-31).

(r.29-31): It is commonly reported that these toxic compounds are frequently detected in poultry feed, constituting a great threat to the health of both animals and humans [2], [5],[10].

Line 53: The term "BW" should be expanded for clarity (body weight) for the first time

Answer: Thank you for your attentiveness. Body weight has been added (r.63-64) to clarify the abbreviation 'BW'.

(r.63-64): Meanwhile, enrofloxacin was found to inhibit CYP3A activity in chickens at doses of 25 and 125 mg/kg body weight (BW) [33].

Line 95: "Focussed" should be "focused"

Answer: Thank you for your remark. The word has been revised.

Line 116: You should specify that ACE is the hydroxylated form of phenacetin (PH).

Answer: Thank you for your helpful comment. The information was added to lines 102-106.

(r.103-106): In this study, the limits of quantification (LOQs) for 4-acetaminophen (ACE, ACE is the hydroxylated form of PH), hydroxybupropion (OH-BP), 6-hydroxychlorzoxazone (OH-CLN), 4-hydroxytolbutamide (OH-TBD), 7-hydroxycoumarin (OH-CAN), and 1-hydroxymidazolam (OH-MDZ) were 5 ng/mL in CLM.

Line 131: “Michaelis-Menten constant” seems to have a different font size.

Answer: Thank you for your attentiveness. The font size was revised (r. 123).

Line 144: Use a colon instead of a comma. In tables 4 and 5 as well (line 176-180 and 186-190)

Answer: Thank you for your useful suggestions. The comma has been replaced by a colon in the captions of the tables and figures (r.147; r.169-173; r.179-183).

Line 454: To which solvent does the reported gradient refer? A or B?

Answer: Thank you for your useful information. The solvent was B in this study, and information was added in the gradient (r. 419-421).

(r.419-421): The gradient elution program was performed as follows: 0-1.5 min (5%-15% B); 1.5-3.5 min (15%-95% B); 3.5-7.5 min (95% B), 7.5-9.5 min (95%-5% B), 9.5-10.0 min (5% B).  

Additionally, I would include, in the supplemental materials, chromatograms of the probes and of AFB1 and TIA.

Answer: Thank you for your helpful remark. Example chromatograms of TIA and AFB1 and of probes have been added to the supplemental materials.

Figure S1. Example chromatogram of tiamulin in broiler chicken plasma

Figure S2. Example chromatogram of aflatoxin B1 in broiler chicken plasma

Figure S3. Example chromatograms of the different probes of CYP450 enzymes: (A) ccetaminophen; (B) OH-bupropion; (C) OH-chlorzoxazone; (D) OH-tolbutamide; (E) OH-coumarin; and (F) OH-midazolam.

Reviewer 4 Report

Comments and Suggestions for Authors

The submitted work is of very high-quality and suitable for this journal.

I recommend the work for publication and have only mild comments about it.

In the material and methodology section, please indicate how the randomization was performed.

In section 5.4.1. Pharmacokinetic study of AFB1 with or without TIA 466 please state what feed was used, that is, the name and manufacturer.

Author Response

Reviewer 4

The submitted work is of very high-quality and suitable for this journal. I recommend the work for publication and have only mild comments about it.

In the material and methodology section, please indicate how the randomization was performed.

Answer:Thank you very much for your useful suggestions. Chickens were selected randomly and numbered according to random numbers generated in Excel. The sentence was added below.

r.438-440: Chickens were selected randomly and numbered according to random numbers generated in Excel.

In section 5.4.1. Pharmacokinetic study of AFB1 with or without TIA 466 please state what feed was used, that is, the name and manufacturer.

Answer:Thank you very much for your remark. A commercial broiler grower diet (511, CP feed, China) was used. Lines 436-437 were added to the manuscript.

(r.436-437): A commercial broiler grower diet (511, CP feed, China) was used.

Round 2

Reviewer 2 Report

Comments and Suggestions for Authors

The manuscript was improved and can now be published as it is.